# Isometric spiracular scaling in scarab beetles—implications for diffusive and advective oxygen transport

**Julian M Wagner[1], C Jaco Klok[1], Meghan E Duell[1], John J Socha[2], Guohua Cao[3], Hao Gong[4], Jon F Harrison[1]\***

[1]School of Life Sciences, Arizona State University, Tempe, United States; [2]Department of Biomedical Engineering and Mechanics, Virginia Tech, Blacksburg, United States; [3]School of Biomedical Engineering, ShanghaiTech University, Shanghai, China; [4]Department of Radiology, Mayo Clinic, Rochester, United States

**Abstract** The scaling of respiratory structures has been hypothesized to be a major driving factor in the evolution of many aspects of animal physiology. Here, we provide the first assessment of the scaling of the spiracles in insects using 10 scarab beetle species differing 180× in mass, including some of the most massive extant insect species. Using X-ray microtomography, we measured the cross-sectional area and depth of all eight spiracles, enabling the calculation of their diffusive and advective capacities. Each of these metrics scaled with geometric isometry. Because diffusive capacities scale with lower slopes than metabolic rates, the largest beetles measured require 10-fold higher $P_{O_2}$ gradients across the spiracles to sustain metabolism by diffusion compared to the smallest species. Large beetles can exchange sufficient oxygen for resting metabolism by diffusion across the spiracles, but not during flight. In contrast, spiracular advective capacities scale similarly or more steeply than metabolic rates, so spiracular advective capacities should match or exceed respiratory demands in the largest beetles. These data illustrate a general principle of gas exchange: scaling of respiratory transport structures with geometric isometry diminishes the potential for diffusive gas exchange but enhances advective capacities; combining such structural scaling with muscle-driven ventilation allows larger animals to achieve high metabolic rates when active.

**\*For correspondence:** j.harrison@asu.edu

**Competing interest:** The authors declare that no competing interests exist.

## Editor's evaluation

This paper is of interest to biologists looking to understand the scaling and geometric constraints of respiratory systems. The work shows that as beetles evolve bigger bodies, their spiracles scale with geometric isometry. As such, advective conductance increases more rapidly with body mass than does their metabolic rate, unlike their diffusive conductance, indicating that bulk air flow is required to sustain respiration in larger insects.

## Introduction

As animal species evolve different sizes, many aspects of their physiology and morphology scale disproportionately with one another (allometrically) with consequences for animal behavior, life history, evolution, and diversity (*Bonner, 2006*; *West, 2017*; *Sibly et al., 2012*). A driver of this disproportionality lies in the nonlinear scaling of geometry: doubling the radius of a sphere gives quadruple the surface area and octuple the volume; in a similar way, scaling up a small body plan gives drastically altered ratios of surface area, volume, and body length. Since the challenges associated with changes in body size have a geometric origin, they are ubiquitous. As a result, understanding the mechanisms animals

use to overcome the effects of changes in geometric proportions remains a pervasive, important, and challenging biological problem. Three related aspects of animal function modulated by allometry are scaling of animal metabolic rates, often scaling with mass$^{0.75}$ (**West et al., 1997**; **Gillooly et al., 2016**), limits on the maximal body sizes of specific taxa (**Kaiser et al., 2007**; **Lane et al., 2017**), and gas exchange strategies (**Perry et al., 2019**). For gas exchange, volume of tissue and hence potential gas exchange needs of animals scale with the cube of length (like the sphere), while surface areas tend to scale with the square of length. This leads to a decline in the ratio of surface area to volume with size. As a consequence, when animals evolve larger sizes, they may need to adapt the proportions of their respiratory structures or increase the use of advection (bulk flow) to avoid facing limitations based on processes that depend more on surface area such as diffusion.

Limitations on the capacity of larger animals to support oxygen delivery to tissues have been proposed to drive the hypometric scaling of metabolic rates with size, as well as the hypometric scaling of many physiological (e.g., heart and ventilation rates) and behavioral/ecological traits (e.g., territory size, dispersal distance) (**Bonner, 2006**; **West, 2017**; **Sibly et al., 2012**; **West et al., 1997**; **Banavar et al., 2010**). However, competing theories suggest that other factors, such as heat dissipation constraints, nutrient uptake constraints, or performance-safety trade-offs, drive the hypometric scaling of metabolic rates and correlated variables, and that evolutionary adaptations of respiratory systems to size allow animals to match oxygen supply to need regardless of body size (**Glazier, 2014**; **Harrison, 2017**; **Harrison, 2018**; **White and Kearney, 2014**). One important step in resolving this controversy is determining how respiratory structures and mechanisms scale. The vast majority of prior studies of the scaling of gas exchange structures have focused on vertebrates, especially mammals. In contrast, there is relatively limited information on the scaling of gas exchange structures in invertebrates, despite the fact that most animal species are invertebrates (**Gillooly et al., 2016**; **Peters, 1983**). The scaling of the insect respiratory system is of particular interest, as aspects of tracheal system structure have been reported to scale hypermetrically, in contrast to the isometric or hypometric scaling of respiratory structures in vertebrates, supporting the hypothesis that possession of a tracheal respiratory system limits insect body size (**Kaiser et al., 2007**; **Harrison et al., 2010**; **Vogt and Dillon, 2013**; **Harrison et al., 2005**; **Greenlee and Henry, 2009**). Here, we report the first study of the scaling of insect spiracles, the gateway of air into the body and the first step in oxygen delivery from air to tissues, presenting new insight into a key morphological pathway in this most biodiverse clade of terrestrial animals.

Gas exchange usually occurs in a series of steps, often a sequence of alternating diffusive and advective processes. The capacity for a respiratory surface to conduct oxygen (diffusive conductance, $G_{\text{diff}}$) can be described using Fick's law, that is

$$G_{\text{diff}} = \frac{\text{area}}{\text{thickness}} * K, \tag{1}$$

where $K$ is Krogh's diffusion constant for oxygen in the barrier. The diffusive oxygen exchange across the surface ($J_{\text{diff}}$, mol s$^{-1}$) is given by

$$J_{\text{diff}} = G_{\text{diff}} * \Delta P_{O2}, \tag{2}$$

where $\Delta P_{O2}$ is the partial pressure gradient for oxygen across the exchanger. When gas exchange relies on diffusion across a barrier, either $G_{\text{diff}}$ or $\Delta P_{O2}$ must increase to match the increased oxygen demand inherent in a larger body size (a larger relative tissue volume), or oxygen supply will limit metabolic rate. Increases in $G_{\text{diff}}$ may be accomplished by either a decrease in diffuser thickness or increase in area. The $\Delta P_{O2}$ from air to mitochondria can be no greater than atmospheric $P_{O2}$ (approximately 21 kPa at sea level); this biophysical constraint sets an upper limit on the ability of large animals to utilize increases in $\Delta P_{O2}$ to overcome a $G_{\text{diff}}$ that does not increase in proportion to oxygen consumption rate.

The scaling of surface area, barrier thickness, and $\Delta P_{O2}$ for gas exchangers across species of animals varies with clade and developmental stage. In adult vertebrates, the scaling of the passive diffusing capacity of the lung across species scales hypometrically, but matches the scaling of metabolic rates (**Gillooly et al., 2016**). However, the scaling of respiratory morphology differs in endotherms and ectotherms (**Gillooly et al., 2016**), as barrier thickness is constant with size in ectotherms, but increases with size in endotherms. As a consequence, endotherms must scale surface area of the lung more steeply than ectotherms to account for their increased barrier thickness and match the scaling of $G_{\text{diff}}$

to the scaling of metabolic rate. Bird eggs, which rely on diffusion through pores for oxygen, employ a different strategy. Eggs of larger species have relatively thicker shells (scaling with mass$^{0.45}$), increasing barrier thickness with size, likely to mitigate a higher likelihood of mechanical damage (*Ar and Rahn, 1985*). Pore area increases proportionally with shell thickness, so $G_{diff}$ per pore is relatively constant across egg size, and larger eggs have a higher density of pores (*Tøien et al., 1988*). The scaling of the $G_{diff}$ of the shell overall matches the scaling of metabolic rate across species, with both scaling hypometrically (*Ar and Rahn, 1985*; *Tøien et al., 1988*). Pycnogonids (sea spiders) show yet another pattern for the diffusing capacity of their respiratory structures (their legs). Unlike either bird eggs or vertebrate lung membranes, pycnogonid barrier thickness scales isometrically (*Lane et al., 2017*). As in bird eggs, there is an increase in the area-specific diffusing capacity of the leg cuticle of larger pycnogonids, although the morphological basis remains unclear (*Lane et al., 2017*). However, the increases in diffusive conductance of the respiratory exchanger are not sufficient to match increases in metabolic rates with size, so the $\Delta P_{O2}$ across the leg cuticle increases in larger pycnogonid species, which may limit maximal species size in this taxa (*Lane et al., 2017*).

Advective steps in gas exchange can occur using either air or aqueous media and represent a second broad strategy for delivering gases to tissues. The morphological capacity for a structure to transport a fluid by advection, $G_{adv}$, $m^4$ s kg$^{-1}$, can be described from Poiseulle's law,

$$G_{adv} = \frac{area^2}{8* \text{ dynamic viscosity } * \text{ length}}. \tag{3}$$

Given this relationship, the advective transport of oxygen through the structure ($J_{adv}$, mol s$^{-1}$) is given by

$$J_{adv} = G_{adv}^*[O_2]^*\Delta HP \tag{4}$$

where [$O_2$] is the concentration of $O_2$ in the fluid (mol m$^{-3}$), and $\Delta HP$ is the hydrostatic pressure gradient across the structure (kg m$^{-1}$ s$^{-2}$). Some examples in mammals illustrate how morphology scales for structures relying on advection. In mammals, the radius of the aorta scales with mass$^{0.375}$, and the length of the aorta scales with mass$^{0.25}$, suggesting that $G_{adv}$ of the aorta scales with mass$^{1.25}$ (4 * 0.375 − 0.25) (*Holt et al., 1981*). The tracheal–bronchial system is the advective structure for air transport in vertebrates; radius scales with mass$^{0.39}$ while lengths scale with mass$^{0.27}$, suggesting that $G_{adv}$ for mammalian aorta and bronchial systems scale with mass$^{1.29}$ (*Stahl, 1967*). $G_{diff}$ for these same structures, on the other hand, scale as mass$^{0.5}$. Thus, the morphological structures of mammalian respiratory systems seem to scale such that advective capacities increase more than metabolic rates in larger species, while diffusive capacities decline. Of course, mammalian oxygen transport through the bronchial tree and circulatory system is thought to rely on advection regardless of size.

The design of the insect tracheal system is fundamentally different from either the vertebrate respiratory system or that of skin-breathing aquatic invertebrates; and it remains unclear how the components of the system scales. In insects, spiracles provide a (usually) gated opening to an air-filled conduit system that branches through the insect, with oxygen transported in the gas phase to the most distal surface of the tracheoles, with transport then occurring in the liquid phase from tracheole to mitochondria (*Harrison et al., 2013*). Since Krogh's demonstration that diffusion should suffice for oxygen transport in a relatively large Lepidopteran larvae, diffusion has been considered to be an important mechanism of gas exchange in insects (*Krogh, 1920*; *Hetz and Bradley, 2005*). However, many insects, including small ones, supplement diffusion with advection, especially when active (*Harrison et al., 2013*; *Socha et al., 2010*; *Wasserthal et al., 2018*). The spiracles are potentially an important step in insect gas exchange, since they are relatively small (difficult to see by eye in most insects) and yet must sustain all gas flux. It appears that spiracle morphology just matches gas exchange needs at peak metabolic performance with little additional capacity; for example, sealing of just one thoracic spiracle reduces flight metabolic rate in *Drosophila* (*Heymann and Lehmann, 2006*). At present it is not clear whether the size of spiracles should best match $G_{diff}$, $G_{adv}$, or some other physiological capacity. To shed light on this question, we used micro-computed tomography (micro-CT) (*Iwan et al., 2015*) to provide the first interspecific examination of the scaling of spiracles, using 10 species of scarab beetles spanning two orders of magnitude in mass, including some of the most massive extant species (*Figure 1*).

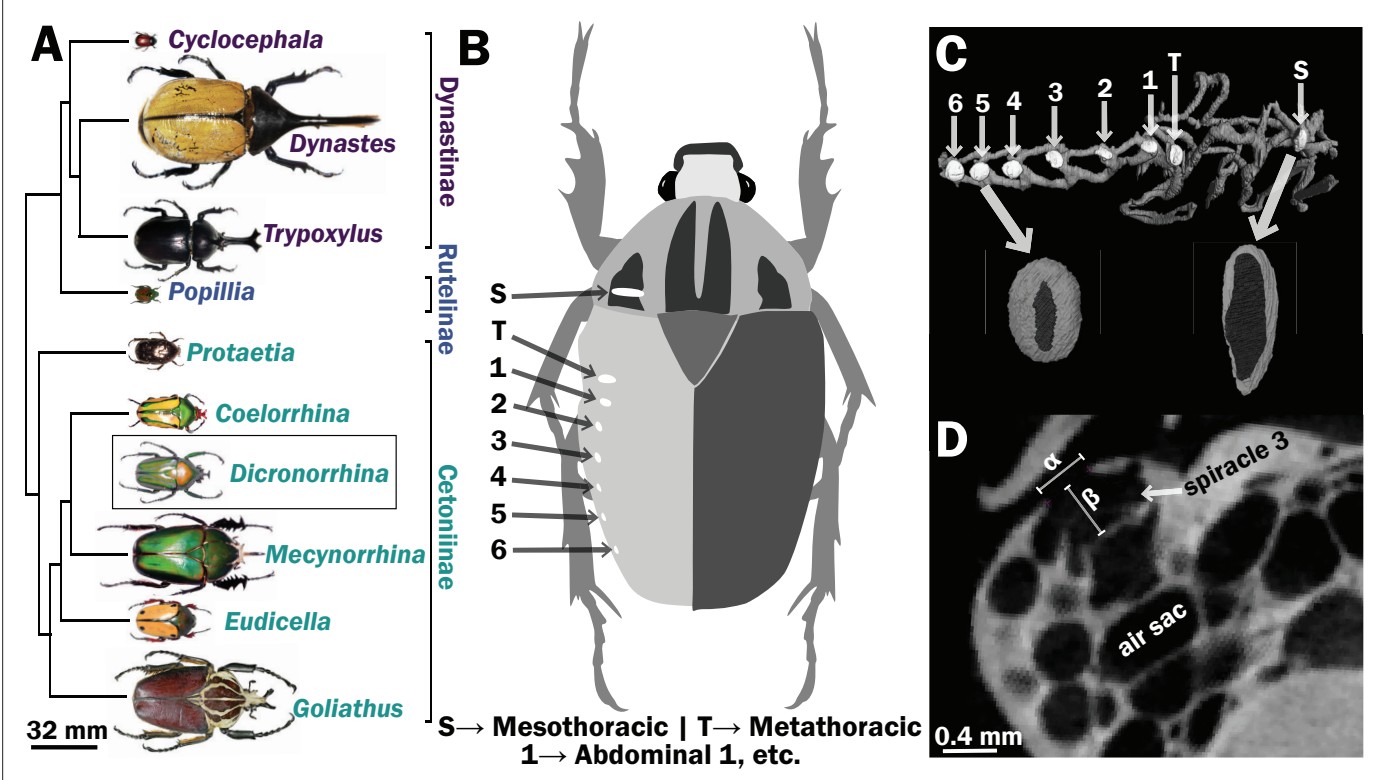

**Figure 1.** Scarab beetles include large bodied individuals and have eight spiracles. (**A**) Phylogenetic tree for the scarab beetles used in this study showing size distribution among clades (branch lengths are meaningless). (**B**) Location of the eight spiracles in the scarab body. (**C**) 3D reconstruction of the tracheal trunks in the thorax, legs, and abdomen of *Dicronorrhina derbyana*; spiracles are shown in white. The larger images of spiracles show the size of the opening (dark in color) compared to the mushroom-shaped (white) atrium behind and the differences in spiracle shape. (**D**) Transverse X-ray slice through the third abdominal spiracle with diameter, α, and depth, β, measures illustrated.

## Results

All spiracles scaled with geometric isometry for area, depth, area/depth (which corresponds to diffusive capacity), and area²/depth (which corresponds to advective capacity) *Figure 1* (*Figure 2*, *Figure 2—figure supplements 3 and 4*, *Supplementary files 1 and 2*). Some example regressions with confidence intervals for the slopes are shown in *Figure 2*, illustrating scaling isometry, the larger size of the mesothoracic spiracle, and the tight size distribution of the more anterior spiracles as compared to the posterior; regressions and confidence intervals for each spiracle are in *Supplementary file 4*. The mesothoracic spiracle was much larger than any of the other spiracles, consistent with the general trend of increasing spiracular area closer to the anterior of the animal (*Figure 2—figure supplement 3*, *Supplementary file 2*). The area of the mesothoracic spiracles was approximately four times larger than both the metathoracic spiracles, and abdominal spiracles 1–3, and abdominal spiracles 4–6 were approximately half the size of the more anterior abdominal spiracles (*Figure 2—figure supplement 3*). Not only were anterior spiracles larger than posterior, but they also had much lower variability around the trend line within the species assayed in this study (*Figure 2*, *Figure 2—figure supplement 3*). In comparison, the depth of the spiracles showed much less variability in tightness of the distribution around the scaling trend lines (*Figure 2*, *Figure 2—figure supplement 3*).

The diffusive capacity of a spiracle ($G_{diff}$, nmol s$^{-1}$ kPa$^{-1}$) at 25°C was calculated using *Equation 1*, with $K$ calculated as $D * \beta$, with $D$ (the diffusivity constant for $O_2$ in air) = 0.178 cm² s$^{-1}$ (*Lide, 1991*) and $\beta$ (the capacitance coefficient for oxygen in air) = 404 nmol cm$^{-3}$ kPa$^{-1}$ (*Piiper et al., 1971*). To calculate total diffusive or advective capacity per beetle, the diffusive/advective capacity for all eight spiracles was summed and doubled (to obtain the total for both sides of the animal). As with individual spiracles, the combined diffusing capacity of all the spiracles scaled isometrically, with a slope not significantly different from 0.33 (*Figure 3A*, *Supplementary files 3–5*). The upper 95% confidence limit for this slope was 0.505, well less than any reported interspecific scaling exponent for metabolic rate.

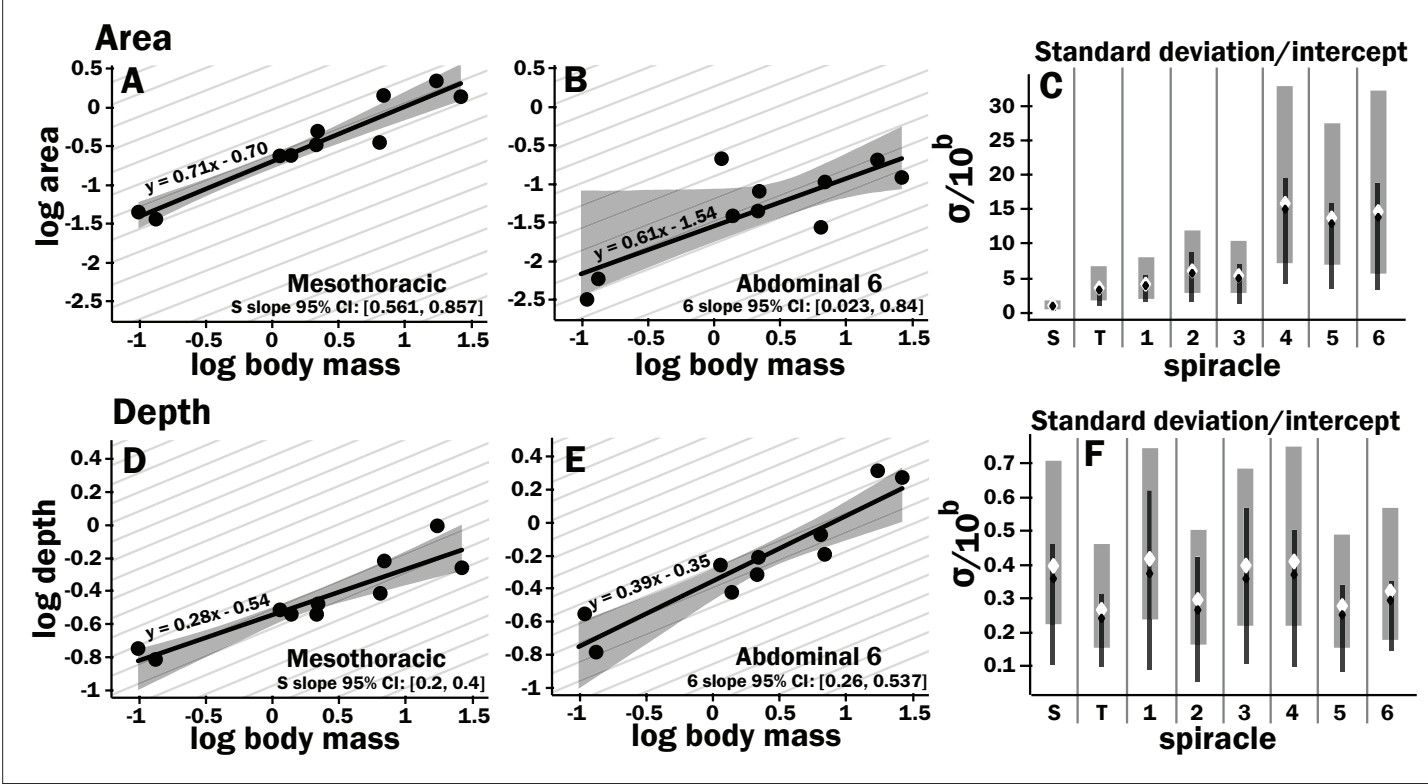

**Figure 2.** Isometric scaling of scarab beetle spiracles. Spiracle area scales with geometric isometry (**A, B**), with much tighter distribution about the isometric model for the large anterior spiracles compared to the smaller posterior spiracles. In A, B, D, and E, the light gray lines show isometric scaling (slopes of 0.67 for area and 0.33 for depth). (**C**) shows estimates for the variability for regression models for the various spiracles (S mesothoracic, T metathoracic, 1–6 abdominal), calculated as the standard deviation divided by $10^{\text{regression intercept}}$, which represents the spiracle area for a 1 g beetle. Black diamond and line show the median and 2.5th–97th residual standard deviation divided by $10^{\text{regression intercept}}$ calculated on nonparametric bootstrap samples. The white diamond and gray interval represent the median and 3rd–97th highest posterior density interval for the standard deviation divided by $10^{\text{regression intercept}}$ calculated from parameter samples from the Bayesian regression. We see a trend toward much higher variability in posterior spiracle area as compared to anterior. In contrast to spiracle area, spiracle depth shows similar variability in all spiracles (**D–F**) regardless of position.

The online version of this article includes the following figure supplement(s) for figure 2:

**Figure supplement 1.** Nonidentifiability of phylogenetic signal parameter in pGLS regression model.

**Figure supplement 2.** Nonidentifiability of the phylogenetic signal parameter as indicated by Bayesian modeling.

**Figure supplement 3.** Slopes, intercepts, standard deviations, and quasi-coefficient of variation with confidence intervals generated via nonparametric bootstrapping or Bayesian regression for all spiracles and area, depth, area/depth, or area²/depth versus mass.

**Figure supplement 4.** Regression plots for both Bayesian and nonparametric bootstrap analyses.

The $\Delta P_{O_2}$ across the spiracles if gas exchange occurs completely by diffusion was calculated for various oxygen consumption rates using ***Equation 2***. To calculate the $\Delta P_{O_2}$ across the spiracles needed to supply the beetle's total resting metabolic demand by diffusion, the metabolic rate for a quiescent beetle at a body temperature of 25°C of a given mass was estimated from ***Chown et al., 2007*** with the following equation: $\log_{10}(\text{metabolic rate } (\mu W)) = 3.2 + 0.75 \log_{10}(\text{mass (g)})$. This metabolic rate was converted to an oxygen consumption rate assuming an RQ of 0.85 (20.7 µj nl⁻¹). For resting metabolic rate (slope of 0.75, shown as the repeated light gray background lines), the required $p_{O_2}$ gradient across spiracles necessary to supply oxygen by diffusion was small but increased by an order of magnitude, from about 0.05 kPa in the smallest beetles to nearly 0.5 kPa in the largest scarabs (***Figure 3B***).

Estimating the scaling of gas exchange during flight of flying beetles has uncertainties. Niven and Scharlemann calculated a scaling coefficient for insect flight of 1.07 (***Niven and Scharlemann, 2005***). Duell and Harrison recently reassessed the scaling of flight metabolism in insects, and found that the scaling coefficients for flight metabolic rates depended on insect size, with a scaling coefficient of 1.19 for insects weighing less than 58 mg and of 0.67 for insects weighing more than 58 mg (***Duell et al., 2022***). As all of the beetles used in this study were larger than 58 mg, it seems likely that their flight

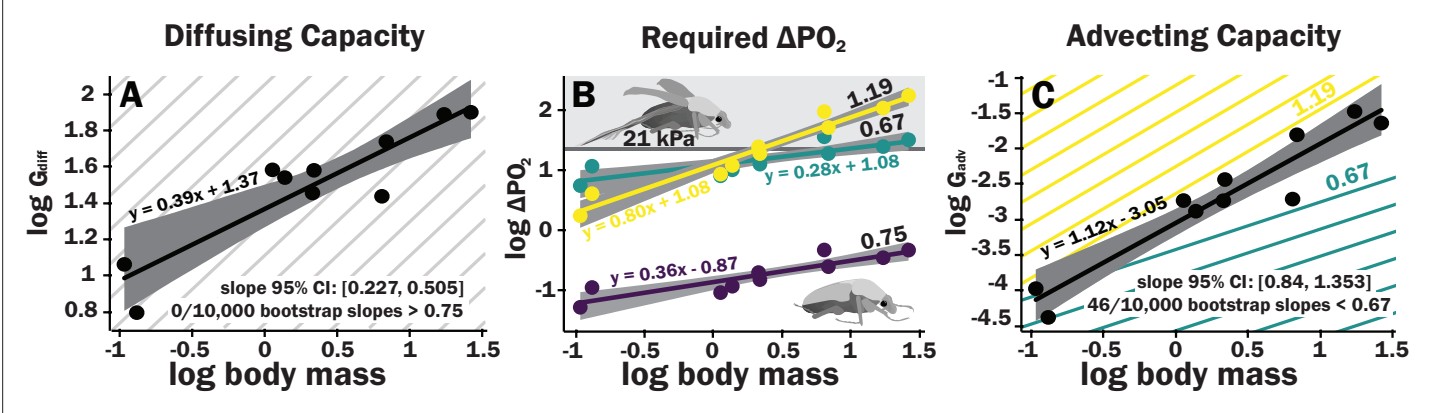

**Figure 3.** Scaling of the spiracles is insufficient for diffusive capacities across the spiracles to match expected increases in metabolic rate, so for pure diffusive gas exchange, the required partial pressure for oxygen must increase with size. In contrast, advective capacities through the spiracles likely match or exceed the scaling of flight metabolic rate. (**A**) The $log_{10}$ of total spiracular diffusive capacity per beetle (nmol $s^{-1}$ $kPa^{-1}$) increases with beetle size, with a slope estimated as 0.39. This slope was not significantly different from the 0.33 predicted from isometric scaling. The upper 95% confidence limit for the slope was 0.505, lower than any reported metabolic scaling slopes for insects. A metabolic rate slope of 0.75, commonly found for resting insects and animals more generally, is shown in light gray. (**B**) The $log_{10}$ $P_{O2}$ gradient (kPa) across the spiracles required to diffusively supply the oxygen demand of beetles increases with beetle size. The lower, purple line shows the estimated $P_{O2}$ gradient across the spiracles to support diffusive gas exchange at rest; this increases from approximately 0.05–0.49 kPa as beetles increase in body size across this range. The less steeply upward sloping greenish and steeper yellow lines shows the estimated $P_{O2}$ gradient across the spiracles during flight, assuming a 90× aerobic scope if flight metabolic rates scale with an exponent of 0.67 or 1.19, as found for large insects and small insects, respectively (**Duell et al., 2022**). The upper gray band indicates where the partial pressure of oxygen needed for calculated beetle metabolic demand exceeds the 21 kPa atmospheric oxygen level. (**C**) Hypermetric scaling of $log_{10}$ summed advective capacity ($m^3$ $s^{-1}$ $kPa^{-1}$) versus $log_{10}$(body mass). There are uncertainties in the scaling of metabolic rate in flying insects: depending on size and study, slopes have ranged from 0.67 to 1.19. Confidence limits for advecting capacity include 1.19 but not 0.67. Equations of regression lines and confidence intervals for the slopes are shown for each plot.

metabolic rates scale hypometrically, with a slope less than 1. Scaling patterns can vary across clades (**Ehnes et al., 2011**; **Capellini et al., 2010**), so ideally, we would employ measures of the flight metabolic rates of the scarab beetles in this study. Unfortunately, most beetles cannot sustain flight in the small containers required for respirometry. Flight metabolic rates have as yet only been reported for four species (**Duell et al., 2022**), and the slope of the scaling relationship for these four species has great uncertainty. Thus, based on the current literature, the scaling exponent flight metabolic rates in insect likely ranges between 0.67 and 1.19, depending on size.

For calculations of the partial pressure gradient across the spiracles during flight, a critical factor is the magnitude of the aerobic scope. Three studies to date measured resting and flight metabolic rates for beetles ranging in body mass from 0.3 to 1.3 g; two used tethered flight and one free flight (**Chappell, 1997**; **Rogowitz and Chappell, 2000**; **Auerswald et al., 1998**). Because it is challenging to induce maximal flight performance and measure aerobic metabolic rate, and we are interested in what the oxygen partial pressure gradient might be across the spiracles during maximal flight performance, we used the highest aerobic scopes reported for individual beetles in these studies, which were 80, 90, and 110× higher than resting metabolic rates. We used the median of these values (90×) to estimate maximal aerobic metabolic rate during flight relative to quiescent one gram beetles. Because there is uncertainty in the scaling of metabolic rates during flight, the required $P_{O2}$ gradient across the spiracles to support gas exchange by diffusion at rest and during flight was calculated by rearranging **Equation 2** and performing unit conversions as follows:

$$\Delta P_{O2}\ (\text{kPa}) = \left( \frac{\left(10^{log_{10}(\text{AS})\ +\ 3.20 + \text{EXP}*log_{10}(\text{mass (g)})} \mu W\right) \left(\frac{1\ \frac{\mu J}{s}}{1\ \mu W}\right) \left(\frac{nL}{20.7\ \mu J}\right) \left(\frac{1\ nmol}{24.5\ nL}\right)}{\left(\frac{\text{area}}{\text{depth}}\ (\text{cm})\right) \left(0.178\ \frac{cm^2}{sec}\right) \left(404\ \frac{nmol}{cm^3 kPa}\right)} \right), \tag{5}$$

where conversion factors of 20.7 kJ/l and 24.5 mol/l were assumed for $O_2$ at 25°C, AS is the aerobic scope (1 for resting [$log_{10}(1) = 0$ in the equation] and 90 for a flying 1 g insect [$log_{10}(90) = 1.954$ in the equation]), and EXP is the scaling exponent for metabolic rate.

For small beetles, the estimated $P_{O22}$ gradient across spiracles during flight was 2–5 kPa. Thus, plausibly, beetles in the smallest size range may be able to deliver sufficient oxygen to the tissues by diffusion, though further studies of conductance of the tracheal system between the spiracles and flight muscles will be required to answer this question. For the largest beetles, the required $P_{O2}$ gradient across spiracles during flight substantially exceeded 21 kPa, indicating that during maximal aerobic flight performance, diffusion cannot supply oxygen across the spiracles, and certainly not to the flight muscles (*Figure 3B*, *Supplementary file 5*). Regardless of whether flight metabolic rate scales with exponents of 0.67–1.19, because metabolic rates increase more with size than spiracular diffusing capacity, the required $P_{O2}$ gradient across spiracles increases with body size for diffusive gas exchange (*Figure 3B*).

Advective capacity was calculated using *Equation 3*, assuming a dynamic viscosity of air of 1.86 × $10^{-8}$ kPa s (*Lide, 1991*). The calculated advective capacity for all spiracles increased with an estimated slope of 1.1, that was greater (95% confidence limits 0.84–1.34) than the minimum scaling exponent for flight metabolic rate (0.67), also greater than the estimated slope of metabolic rate for resting insects (slope of 0.75, light gray lines), but included maximum scaling exponent reported for flight metabolic rates in insects (1.19, *Figure 3C*).

## Discussion

Spiracles scaled with geometric isometry. Isometric scaling of diffusive capacities means that diffusion becomes increasingly less able to meet oxygen demands in larger beetles, with the required gradient for oxygen transport by diffusion through the spiracles increasing by an order of magnitude over two orders of magnitude in body mass. Conversely, our data demonstrate that the advective capacities of the spiracles scale more positively than resting metabolic rates. For flight metabolic rates, uncertainties in the scaling exponent for flight metabolic rates of beetles means that we can only conclude that the scaling exponents for advective capacities of the spiracles may match or possibly exceed those of metabolic rates. These results demonstrate that large insects must rely on advection through the spiracles in order to achieve their maximal aerobic flight metabolic rates. Our results also imply that there is no physical constraint associated with spiracular gas exchange that limits insect size and metabolic rates.

It is important to note that our analysis only assessed required $P_{O2}$ gradient across spiracles, not within the entire tracheal system. Within the body of insects, gases must be transported through the large tracheal trunks, and then down the branching smaller tracheae and tracheoles in the gas phase, and then finally through liquid phases in the ends of the tracheoles and from the tracheoles to the mitochondria. As yet, we have little information on the relative resistances of these various steps. Based on the $P_{CO2}$ gradient between the spiracles and tracheal trunks, and between the tracheal trunks and the hemolymph (perhaps similar to cellular $P_{CO2}$), the resistance of the internal tracheal system to $CO_2$ transport to active muscles likely substantially exceeds that of the spiracles in active, locomoting animals, whereas spiracular and tracheal resistances may be similar in resting animals (*Harrison, 1997*). This raises the interesting question of whether large insects can supply their resting metabolic rate by diffusion alone. The fact that the calculated required $P_{O2}$ gradient across the spiracles required to sustain resting metabolic rate for the largest beetles in this study is only 0.5 kPa would suggest that the answer is yes. This conclusion is supported by our experience (unpublished observations) that even very large larval and adult scarab beetles (>30 g) can recover from anoxia, which strongly suggests that diffusion can sustain at least the minimal aerobic metabolic rate necessary to restart ventilation.

While the spiracles scale isometrically in beetles, this pattern does not occur universally for tracheal structures, or consistently across clades. Comparing tenebrionid beetles interspecifically, the leg tracheae scale hypermetrically, but the head tracheae scale isometrically (*Kaiser et al., 2007*). Within a bumblebee species, one spiracle scales isometrically (*Vogt and Dillon, 2013*). In the leg of growing locust (*Schistocerca americana*), the diffusing capacity of the large longitudinal tracheae of the leg scales hypometrically (*Harrison et al., 2005*), whereas in a growing caterpillar (*Manduca sexta*), diameters of most tracheae scale isometrically (*Lundquist et al., 2018*). Why different scaling patterns are observed in these different cases is unclear; more in-depth analysis of the required gas transport and the mechanism of transport are needed to evaluate the scaling of individual tracheal system structures. Plausibly the various steps in gas exchange scale similarly (a hypothesis of symmorphosis) as

has been suggested for mammalian respiratory systems (*Weibel et al., 1991*), but resolution of this question in insects will require further study.

Diffusive capacities of the spiracles scaled with mass$^{0.39}$, well below the scaling slope for resting oxygen consumption rate (approximately 0.75); thus, diffusion across the spiracles becomes more challenging for larger insects. The required $O_2$ gradient across the spiracles to supply the metabolic demand by diffusion increases by approximately an order of magnitude from our smallest to largest beetles, but the size effect on the required $P_{O_2}$ gradient is less important than the effect of activity. For quiescent beetles, the $P_{O_2}$ gradients across the spiracles necessary for diffusion are low (0.05–0.5 kPa depending on size). However, during endothermic flight, the required $P_{O_2}$ gradient across the spiracles increases from 5 to 35 kPa for a scaling exponent of 0.67 or from 2 to 174 kPa for a scaling exponent of 1.19, which is impossible to achieve because the maximum partial pressure of oxygen in air is only 21 kPa. With metabolic rates scaling with mass$^{0.75}$ and spiracular depth with mass$^{0.33}$, spiracular area would need to scale with mass$^{1.08}$ (0.75 + 0.33) to conserve the required $P_{O_2}$ gradient to support diffusion across all insect sizes.

By contrast, advective capacity scales with mass$^{1.1}$, exceeding or matching the scaling of flight metabolic rate, depending on insect size (*Duell et al., 2022*). Interpretation of these results is challenging as we do not know how ventilatory flow varies with size in insects. Ventilatory airflow is difficult to measure in insects, because there are so many spiracles, because these spiracles can be variably gated, and because flow can be tidal or unidirectional. If insects can match ventilation to flight metabolic rate across body size, then no changes in oxygen extraction efficiency will be necessary. If ventilatory airflow is matched to flight metabolic rate, then the scaling of advective capacities of the spiracles with an exponent of 1.1 implies that the pressures required to drive convection either remain the same with size (if flight metabolic rate scales with an exponent near 1.1) or falls with size (if flight metabolic rate scales with an exponent of 0.67 or 0.75), for example. Any conclusions on this topic must be very cautious because in the beetles that have been best studied, ventilation during flight includes both tidal flow through some of the thoracic spiracles and unidirectional flow out the abdominal spiracles (*Miller, 1966*). The much larger size of the thoracic than abdominal spiracles suggests that ventilation of the flight muscles may be primarily tidal through the thoracic spiracles, with unidirectional flow out the abdominal spiracles supplying other parts of the body.

We also observe much tighter distributions in the scaling pattern for the area of the large anterior spiracles as compared to the smaller posterior ones. This result may suggest that the large anterior spiracles are more constrained in their morphology, since they presumably provide the gas exchange needed for metabolically demanding tissues like the flight muscle.

There are some important caveats when interpreting our data. Insect spiracles are morphologically complex structures. We made 3D measurements using tomographic imaging, but analyzed air transport capacities by modeling the spiracle as a cylinder, which could over or underestimate capacity depending on factors like valve position and the complex shape of the spiracular atrium. Furthermore, our CT scans were conducted on sacrificed specimens; the assessment of spiracles of living insects could offer insights not possible with static morphology. For example, living insects might control the shape of the bellows-like atrium and valves in a concerted way to promote air flow. As yet, we know little about how the tracheal system structure and function might scale differently in different species. As an example of a fairly dramatic difference in tracheal system function across beetle clades, some Cerambycid beetles use draft inward ventilation through the mesothoracic spiracle during flight, whereas most scarab beetles autoventilate the thorax using wing movements (*Miller, 1966*; *Amos and Miller, 1965*). Dung beetle species vary between exhaling nearly all to none of their air out the mesothoracic spiracles, with species from more arid environments exhibiting more expiration via the mesothoracic spiracle (*Duncan and Byrne, 2005*). Multiple beetle species collapse parts of their tracheal system to produce advective airflow, both in adults and as pupae (*Pendar et al., 2015*; *Socha et al., 2008*; *Waters et al., 2013*). Though it is unclear how respiration via tracheal collapse differs with size and in different species, the prevalence of active breathing further highlights the need for advective airflow for insect function. The phylogenetic, life history, and environmental influences on tracheal system structure, function, and scaling seem likely to be a ripe area for future research.

Our finding of isometric scaling of insect spiracles would appear to differ from reports for tracheae of mammals, in which radius scales with mass$^{0.39}$ and lengths with mass$^{0.27}$ (*Tenney and Bartlett, 1967*). However, confidence limits from our study included these scaling slopes, suggesting that respiratory

scaling of tracheal morphology may be congruent across these disparate groups. Tenney and Bartlett's study (**Tenney and Bartlett, 1967**) had greater power, as it examined 43 species ranging over 5 orders of magnitude in body mass. However, it worth noting that they did not consider error in their slope estimates, test for statistical differences in slopes between the radii and lengths, or consider phylogeny, so the conclusion that mammalian tracheae scale nonisometrically (and differently from insect spiracles) could benefit from rigorous comparative analysis.

Conclusions: Insect spiracles scale with geometric isometry in beetles, which means that diffusive capacities increase much less than metabolic rates as body size increases, while advective capacities increase similarly or more rapidly than do metabolic rates. These are general principles of gas exchange that should apply to respiratory structures of any animal clade exhibiting isometric scaling. For resting insects, the required $P_{O_2}$ gradient across the spiracles necessary to supply resting oxygen consumption increases strongly with size, but remains small in even the largest insects, suggesting that resting gas exchange can be accomplished by diffusion even in very large insects. In contrast, our data clearly demonstrate that maximal aerobic flight cannot be accomplished by diffusion in large beetles.

## Methods
### Acquisition of raw micro-CT images
Seventeen individuals of 10 species (1–2 individuals per species) of scarab beetles (**Figure 1a**) with a size range from 0.097 to 18 g were obtained via breeders from online sources. We examined the following species: *Goliathus goliathus*, *Coelorrhina hornimani*, *Dicronorrhina derbyana*, *Mecynorrhina torquata*, *Eudicella euthalia*, *Protaetia orientalis*, *Popilia japonica*, *Trypoxylus dichotomus*, *Dynastes hercules*, and *Cyclocephalis borealis*. Most species had both male and females represented. Most specimens were scanned using a micro-CT scanner (Skyscan 1172, Bruker, Bilerica, MA, USA) equipped with a Hammamatsu 1.3 MP camera and Hammamatsu SkyScan Control software at Virginia Tech. To maintain tracheal structure in their natural configuration, we used a minimal preparation of fresh samples (**Socha and DeCarlo, 2008**). All beetles were killed using ethyl acetate fumes, stored at 4°C, and scanned within 3 days. They were warmed back to room temperature to avoid motion artifacts from fluid flow, placed in X-ray translucent polyimide tubing (Kapton, Dupont), and centered on a brass stage with putty. Power was set at 10 W, voltage was adjusted for optimum brightness and contrast (70–96 kV), with currents between 104 and 141 µA. Beetles were scanned with 0.4° rotation steps for 180° with frame averaging. A flat-field correction was applied to all scans to account for subtract aberrations. All raw projection images were collected a size of 1024 × 1280 pixels, yielding a scaling of 12–98 µm/pixel that was independent of beetle size. Average measured spiracle dimensions for width/height/depth for the smallest beetles were around 10 pixels and hence resolvable with minimally 10% resolution. Small beetles could be captured in a single scan, but larger beetles were scanned sequentially in segments along their longitudinal axis by varying their position relative to the beam.

*Dynastes hercules* were too large to be scanned with the same instrument, so these beetles were imaged using an in-house-built bench-top micro-focus X-ray computed tomography (micro-CT) platform (see **Sen Sharma et al., 2014**; **Gong et al., 2015** for details) at Virginia Tech. The X-ray tube (Oxford Instrument, Inc) was operated at 70 kV and tube power was fixed at 20 W. Images were collected with an X-ray flat-panel detector (model C7921, Hamamatsu, Inc) operated at 1 × 1 binning mode, with a detector element size of 50 × 50 µm. The axial scanning field-of-view was 37.2 mm in diameter. In each scan, images were collected at 0.5° intervals as the beetle was rotated through 360°, resulting in a total of 720 X-ray projections per scan. Because the specimen was larger than the field of view, multiple scans were conducted sequentially along the animal's anterior–posterior axis to image the entire body. The axial slice images were reconstructed using the standard filtered back-projection reconstruction algorithm, with an image matrix of 1008 × 1012 px and an isotropic pixel size of 36.8 × 36.8 µm.

### Image reconstruction and measurements
Raw micro-CT images were imported into NRecon reconstruction software from SkyScan (Bruker, Bilerica, MA, USA). Ring artifact and beam hardening corrections were applied where necessary,

and contrast was optimized using the software's interactive histogram feature. For large beetles that required multiple scans, reconstructions were set to align and fuse automatically. Slices generated in NRecon were imported into Avizo 9 software (Thermo Fisher Scientific, Waltham, MA, USA) for 3D reconstructions.

Spiracles were identified by the characteristic slit-like shape of the opening, and the bellows-shaped air sac behind it (*Figure 1B–D*). Spiracle locations were confirmed by dissection on representative specimens. Measurements were taken for one of the paired six abdominal and two thoracic spiracles for each beetle (*Figure 1B, C*). A few scans had small aberrant regions (e.g., blurriness) due to challenges in scanning, so which spiracle was measured varied between the symmetric right and left side of an animal based on which region of the scan was best resolved. Diameters of the spiracular opening were measured at the widest point of opening to the outside air in the transverse and sagittal planes. Area of the opening was then calculated assuming an elliptical shape, with the lengths of the semi-major/minor axes being the diameters measured in the transverse and sagittal planes (*Figure 1D*). The depth of the spiracle was measured from the outer opening to the interior valve connecting the spiracle to the tracheal trunk (*Figure 1D*). The sex, mass and dimensions of all measured spiracles are provided in *Supplementary files 1 and 2*.

## Calculations and statistical analyses

We measured the scaling relationships for each spiracle separately, using log–log plots. As dependent variables in these regressions, we tested $\log_{10}$ transformed spiracular depth, area, area/depth (as an index of the diffusive capacity of the spiracle, see *Equation 1*), and area$^2$/depth (as an index of the resistance of the spiracle to advective flow, see *Equation 3*).

We used two statistical approaches to assess the role of the phylogenetic relatedness of the animals in scaling patterns: a phylogenetic generalized linear model (pGLS) and a generative Bayesian model. We ran and plotted pGLS results in R (*Garnier, 2016*; *Orme, 2018*; *Pinheiro et al., 2021*; *Harmon et al., 2008*; *Revell, 2012*; *Paradis et al., 2004*; *R Development Core Team, 2016*). The goal of pGLS is to account for nonindependence of data points due to phylogenetic relatedness in construction of the linear model, which requires a phylogeny of the study species (*Freckleton et al., 2002*). We spliced together such a phylogeny from multiple published scarab phylogenies. The branch positions for beetle subfamilies (Dynastinae, Rutelinae, and Cetoniinae) were determined using *Hunt et al., 2007*. The branches within Dynastinae were placed in the tree using work from *Rowland and Miller, 2012*, and the branches of Cetoniinae determined with two trees, one from *Holm, 1993*; *Micó et al., 1993* and the other Holm (*Holm, 1993*; *Micó et al., 1993*). Four of the genera in this study were present in the tree for Coleoptera constructed by *Bocak et al., 1993*, which indicated the same branch places as our spliced tree, providing some positive confirmation for this tree structure. Branch lengths were set to a value of one because actual branch lengths are not known. Similar to pGLS, we built a Bayesian model assuming that the data were generated by a multivariate normal distribution with the covariance matrix given by the amount of shared ancestry between species (amount of shared branch length). See supplemental methods for the details of the model, selection of priors, and python code. Detailed information on the analysis is available at the website for the paper here: https://julianmwagner.github.io/spiracle_scaling/ and at the corresponding repository (https://github.com/julianmwagner/spiracle_scaling, copy archived at swh:1:rev:0ad9383b23d156430ad-caae2d53861b595205e72; *Wagner, 2022*). Analyses indicated that the parameter characterizing the degree of phylogenetic signal in our data ($\lambda$) was nonidentifiable (*Figure 2—figure supplements 1 and 2*); this result means that our data do not inform this parameter and it could take on any value from zero (no phylogenetic signal) to one (strong phylogenetic signal) with similar probability. Hence, we opted to omit the use of phylogenetic covariance from our models since (1) the total nonidentifiability made selecting a single $\lambda$ via maximum likelihood for the frequentist pGLS dubious, and (2) including it added no explanatory value to our Bayesian regression (parameter samples for $\lambda$ were essentially straight from the prior). We instead used nonparametric bootstrapping (10,000 bootstrap replicates with ordinary least squares regression slopes/intercepts/residual standard deviation as the summary statistics) to obtain confidence intervals for our slope and intercept values. Additionally, we performed a Bayesian linear regression. Our model was a normal likelihood with mean given by a line with slope and intercept parameters. To obtain parameter estimates, we sampled using the Stan implementation of Hamiltonian Monte Carlo (cmdstanpy) in Python. See supplemental methods for

the details of the model, selection of priors, and Python code and at the paper website/repository listed above. No data were excluded.

We defined isometric scaling as scaling as follows: $mass^{0.67}$ for areas $mass^{0.33}$ for area/depth $mass^{1}$ for $area^2$/depth, according to basic principles of geometric similarity (assuming mass is proportional to volume). We observed whether the 95% confidence interval given by bootstrapping/parameter samples for the slope of our measures of spiracle morphology overlapped with the isometric prediction. To produce any p values, we calculated the number of bootstrap replicates with test statistic at least as extreme as a particular value of interest, for example slope compared to isometry.

## Acknowledgements

This research was supported in part by funds from the School of Life Sciences Undergraduate Research (SOLUR) Program through the School of Life Sciences at Arizona State University, Tempe Campus, and by NSF IOS 1122157 and 1558052.

## Additional information

### Funding

| Funder | Grant reference number | Author |
|---|---|---|
| National Science Foundation | IOS 1122157 | Jon F Harrison |
| National Science Foundation | IOS 1558052. | John J Socha<br>Jon F Harrison |

The funders had no role in study design, data collection, and interpretation, or the decision to submit the work for publication.

### Author contributions

Julian M Wagner, Investigation, Methodology, Validation, Visualization, Writing – original draft, Writing – review and editing; C Jaco Klok, Writing – original draft, Methodology, Writing – review and editing, Visualization; Meghan E Duell, Methodology, Writing – review and editing, Visualization; John J Socha, Resources, Funding acquisition, Investigation, Methodology, Supervision, Writing – review and editing, Visualization; Guohua Cao, Resources, Supervision, Visualization; Hao Gong, Software, Supervision; Jon F Harrison, Conceptualization, Resources, Data curation, Software, Writing – original draft, Funding acquisition, Methodology, Writing – review and editing, Validation, Project administration, Visualization

### Author ORCIDs

John J Socha http://orcid.org/0000-0002-4465-1097
Jon F Harrison http://orcid.org/0000-0001-5223-216X

### Decision letter and Author response

Decision letter https://doi.org/10.7554/eLife.82129.sa1
Author response https://doi.org/10.7554/eLife.82129.sa2

## Additional files

### Supplementary files

- Supplementary file 1. The measurements made on each spiracle for each beetle.

- Supplementary file 2. Area and depth measurements for each spiracle for each beetle.

- Supplementary file 3. Conductances for diffusion and advection for each spiracle for each beetle.

- Supplementary file 4. Intercepts and slopes for scaling relationships (log–log plots) for morphologies (spiracular areas and depths), conductances, and $P_{O2}$ gradients across the spiracles, including confidence limits, for each spiracle and for all spiracles combined.

• Supplementary file 5. Summed spiracular areas and average spiracular depths, summed conductances, and estimated average $P_{O_2}$ gradients across all the spiracles, for each beetle species, estimated at rest and during flight if flight metabolic rate scale with mass$^{0.67}$ or mass$^{1.19}$.

• Supplementary file 6. Reconstruction of the spiracles and connecting tracheal trunks on one side of the scarab beetle *Dicronorhina derbyana*.

• MDAR checklist

## Data availability

All data are provided in the supplementary files.

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
