## [Editor Report]

This paper is of interest to biologists looking to understand the scaling and geometric constraints of respiratory systems. The work shows that as beetles evolve bigger bodies, their spiracles scale with geometric isometry. As such, advective conductance increases more rapidly with body mass than does their metabolic rate, unlike their diffusive conductance, indicating that bulk air flow is required to sustain respiration in larger insects.

---

## [Decision Letter]

**Decision letter after peer review:**

[Editors’ note: the authors submitted for reconsideration following the decision after peer review. What follows is the decision letter after the first round of review.]

Thank you for submitting the paper "Isometric Spiracular Scaling in Scarab Beetles: Implications for Diffusive and Advective Oxygen Transport" for consideration by *eLife*. Your article has been reviewed by 2 peer reviewers, and the evaluation has been overseen by a Reviewing Editor and a Senior Editor. The following individual involved in review of your submission has agreed to reveal their identity: Philip GD Matthews (Reviewer #1).

Comments to the Authors:

We are sorry to say that, after consultation with the reviewers, we have decided that this work will not be considered further for publication by *eLife*.

That said, because there are otherwise many positives in your work, I would sincerely welcome a new submission if the major concerns noted by the reviewers – related to stronger justification/support for the scaling assumptions on which the results are dependent – could be addressed adequately, along with your best effort to address other points also raised in the reviews below.

*Reviewer #1 (Recommendations for the authors):*

The authors were examining how the morphology of insect spiracles change as insects increase in body mass, specifically to determine how their diffusive and advective conductances scale with body mass, and how these exponents compare with the exponent describing the scaling of metabolic rate with body mass.

The scaling analysis uses spiracle morphology extracted from 3D micro-CT scans of beetles taken across a 180-fold range of body mass. This approach clearly reveals that spiracles scale with exponents that are not significantly different from those predicted from geometric isometry, with linear dimensions increasing with body mass (M) as M^0.33, area M^0.67, and volume M^1. From these geometric relationships the diffusive and advective conductance of the spiracles were estimated using reasonable simplifying assumptions. Clearly, diffusion cannot deliver sufficient oxygen to sustain the metabolic requirements of large or active beetles, and the bulk flow of air must be required. Given that the advective conductance scales with M^1.1, this mechanism scales with a sufficiently high exponent meet the oxygen delivery needs of active insects of all masses.

Comparing the scaling exponents for spiracle conductance against an exponent of 0.75 for insect metabolic rate is potentially an issue, since the capacity of the respiratory system and its spiracles should be matched to the insect's maximum aerobic O2 uptake requirements (i.e., during flight or activity) not their metabolic rate at rest when oxygen demand is low. The metabolic rate of flying insects scale with body mass with an exponent that is substantially higher than M^0.75, with interspecific analysis suggesting M^1.10 (Niven and Scharlemann 2005) or intraspecifically M^1.02 in hopping locusts (Snelling et al. 2011). Interestingly, the flight exponent is also the same exponent calculated in this study for the scaling of spiracle advective conductance (M^1.1). As such, estimating the metabolic rate of flying endothermic beetles to be 90× resting MR, while also assuming resting metabolic rate scales with mass^0.75, doesn't capture the iso/hyperallometric relationship between body mass and metabolic rate during flight or vigorous locomotory activity, but the scaling exponent for advective conductance is comparable to the hyperallometric value for flight metabolic rate.

An implicit assumption in this analysis is that how the spiracles scale with body mass reflects the oxygen delivery capacity of the insect's entire respiratory system. This would be true if either every part of the insect's respiratory system has a conductance that is matched with every other part (symmorphic), such that every part of the gas exchange pathway shared the same scaling exponent and equal capacity for oxygen delivery. If not, then the spiracles must possess the lowest (i.e., rate-limiting) conductance for oxygen uptake – the rest of the tracheal system could have an overall higher capacity or a higher scaling exponent, but this capacity would be unrealized due to the restricted transport of oxygen through the spiracles. In either of these cases, the scaling relationship of the spiracles conductance with body mass would be the same as the oxygen delivery capacity of the insect's tracheal system. However, it is also possible that spiracular conductance scales with a higher exponent than the conductance of other parts of the respiratory system, in which case the spiracles might not be the rate-limiting part of the oxygen transport pathway, and would possess an unrealized capacity. In this case, the scaling exponent for the spiracles would not reflect the tracheal system's oxygen delivery capacity. The validity of the authors' choice in assuming that the insect respiratory system is symmorphic (or that the spiracles are rate-limiting) is partly addressed with the observation that the metabolic rate of *Drosophila* is reduced when a single thoracic spiracle is blocked (L124). But is there any other evidence to support symmorphosis or rate-limiting spiracles? Given that this relationship must be true to accept the overall argument being presented, it would be good to see additional arguments put forward to support this position.

How the conductance of the spiracles is related to the insect's total oxygen uptake is even more complicated when considering advection, since oxygen delivery through the tracheal system isn't limited by atmospheric oxygen partial pressure as diffusion rate is, and can be increased dramatically, essentially only being limited by the capacity of the tracheal air-sacs' ability to pump air. However, how the tracheal pump's power (Power = Pressure x Flow or 'Q') scales with insect body mass may not yet be known. Without knowing this, there is some uncertainty in predicting what the oxygen delivery associated with the hyper-allometric scaling of spiracular advective conductance would be.

While diffusion can operate through all spiracles simultaneously, the total advective capacity can vary depending on which spiracles act as influx and efflux points (assuming continuous advective flow), as well as how these elements are interconnected, or whether all spiracles function together simultaneously during a period in inhalation followed by exhalation. Thus, the total advective capacity and the resulting oxygen delivery rate is determined by how the spiracles operate together. In this paper the total advective capacity is assumed to be the capacity of all spiracles functioning simultaneously (i.e., the summed capacity of all sixteen spiracles) which would only be possible for half the time (assuming inhalation and exhalation are of equal duration).

Overall, this paper does achieve its aim of generating a valuable data set and using this to determine how spiracle morphology scales with beetle body mass. The analysis presented convincingly shows that while diffusion could support the metabolic oxygen demands of a small or resting beetle, advection is required to deliver the oxygen needed for any energetic activity. This also suggests that an insect's size is not constrained by spiracular gas exchange even if the spiracles grow proportionally with body mass and insect size. From an evolutionary point of view this is interesting as it suggests prehistoric giant insects would likely have conformed to this pattern, indicating their size was not limited by oxygen delivery capacity.

While the morphometrics and scaling exponents that are derived from them are all very nicely done and very clear, I think that there needs to be some discussion to explain the rationale behind your use of the M^0.75 relationship for MR as the exponent you are comparing the spiracular conductance exponents against, rather than using scaling exponents derived for flying or active MR, where the exponent is >1. Likewise, I'd be keen to see some mention of the assumptions underlying why examining the diffusive and advective conductances of the spiracles is revealing, when the capacity of this comparatively small part of the gas exchange pathway may exceed the internal conductance. Ideally more compelling evidence should be provided showing either symmorphosis of gas transport across the tracheal system or that the spiracles are likely to be the rate-limiting conductance within the tracheal system.

Likewise, it'd be great to see some rationale behind how you might expect the tracheal pump's capacity to scale with body mass, since this will determine if the oxygen delivery capacity associated with the spiracle's advective conductance also scales with M^1.1. For example, if it is assumed that tracheal pump power scales isometrically (M^1), then as advective conductance scales with M^1.1, would this increase flow, and therefore oxygen delivery, with the same exponent? Would pressure decrease in larger insects? Presenting some background to the assumptions around how the insect generates an advective flow through its spiracles, and how this might scale with insect body mass, is important to be able to appreciate how increasing spiracle conductance would change the volumetric flow of air and, therefore, oxygen delivery.

It is interesting that the mesothoracic spiracles show the tightest relationship with bodymass, given that these spiracles lie closest to the most metabolically demanding tissue: the thoracic flight musculature. Given the possibility/likelihood of unidirectional advective flow during activity (in through the thoracic and out through the abdominal spiracles), how does the summed advective conductance of the thoracic spiracles compare to that of the summed abdominal spiracular conductance? Is there an excess advective capacity in the abdominal spiracles, assuming they are functioning as "exhaust spiracles" relative to the thoracic "intake spiracles"? Would assuming continuous unidirectional flow (in through some spiracles and out through others) alter the scaling exponent or only the elevation of the advective conductance relationship? Could this be considered in the analysis?

Specific comments:

L33: I'd consider changing the exponent you consider from the resting metabolic rate (M^0.75) to that for flight MR (M^1.1)

L114: "… remains unclear how the components of the system scales". Change to "scale"

L266: "The mesothoracic spiracle was" change to "were"? I know you only measured one, but there are two of them

L294: "one spiracle scales isometrically" change to "one spiracle pair scales isometrically"

*Reviewer #2 (Recommendations for the authors):*

The study aimed to determine gas transport capacity of tracheal spiracles in different sized scarab beetles using micro-CT scans. The authors assumed that metabolic rate scales with a scaling exponent of 0.75. They found that spiracle size does not sufficiently increase with increasing body size to allow diffusive oxygen supply but increases more than required to satisfy metabolic demands during advective gas exchange. The data are of interest for Biologists working on the respiratory system of animals but need experimental proof of the scaling exponent used as a reference.

The entire conclusion of the study is based upon the assumption that metabolic rate exactly scales with a 0.75 exponent. Many previous studies, however, showed that this scaling exponent is only valid among a large range of body sizes and (to some extent) including also vertebrates. In single clades, scaling exponents may significantly be different from 0.75. This means that the finding that diffusion is not sufficient in larger beetles depends on the correct scaling coefficient for metabolic rate in these animals. The authors do not provide separate measurements of metabolic rate to more reliably estimate the 0.75 coefficient in scarab beetles. This is, however, critical for the outcome of the study. In equations 1 and 2, the authors nicely explain that diffusion should linearly depend on spiracle geometry. This assumption matches the data in figure 2, showing slopes close to 0.75. In figure 3A, by contrast, total diffusive capacity increases much less than spiracle geometry, which runs apparently counter to the data in Figure 2. This needs an explanation.

The authors leave open the question of how important spiracle opening area is for oxygen flux compared to the rest of the tracheal system. Even assuming that spiracle area satisfies oxygen supply via diffusion, an animal might rely on advective flow because of other tracheal constraints. The above concern also holds for the slope assuming advective oxygen supply. For very small beetles, moreover, equations 3 and 4 might be too simplistic because they do not consider the fluid mechanic effects associated to flows at low Reynolds number. While Reynolds number-dependent phenomena do not change much at large Reynolds number, the thick boundary layer might hinder advective flow at low Reynolds numbers.

The study determined gas transport capacity of tracheal spiracles in different sized beetles using micro-CT scans. The authors found that spiracle area does not sufficiently increase with increasing body size for diffusive oxygen supply. Assuming advection, by contrast, spiracle area increases more than required to satisfy metabolic demands. The manuscript is written clearly and the topic is of interest for Biologists working on the respiratory system of animals. Although I much sympathize with the approach and the data, my impression is that findings and conclusion are too controversial and thus recommend publication in a more specialized journal.

The authors only compare their findings to the 0.75 slope at resting metabolic rate. On page 7, however, they mention that spiracle morphology should match gas exchange needs at peak metabolic performance. I assume that all tested species are capable of flight (?). As flight costs increase with decreasing body size due to viscous drag on wings and body, we would not expect isometric scaling of spiracle openings for diffusive gas exchange. This aspect should be considered in a revised version of the manuscript.

Length. The manuscript consists of 5 pages Introduction, 7 pages Methods, 1 page Results and 4 pages Discussion sections and thus needs a major revision towards balanced section length. The data set is comparatively small and I suggest to add measurements of metabolic rates for each beetle (see comment above).

Statistics. The data in figures 2 and 3 are barely normally distributed and my impression is that the slopes thus strongly depend on the two data points of the smallest beetles (-1 body mass). As the slope only depends on 10 data points in total, I recommend further statistics that evaluates the unequal(?) data distribution.

---

## [Author Response]

[Editors’ note: the authors resubmitted a revised version of the paper for consideration. What follows is the authors’ response to the first round of review.]

Reviewer #1 (Recommendations for the authors):The authors were examining how the morphology of insect spiracles change as insects increase in body mass, specifically to determine how their diffusive and advective conductances scale with body mass, and how these exponents compare with the exponent describing the scaling of metabolic rate with body mass.The scaling analysis uses spiracle morphology extracted from 3D micro-CT scans of beetles taken across a 180-fold range of body mass. This approach clearly reveals that spiracles scale with exponents that are not significantly different from those predicted from geometric isometry, with linear dimensions increasing with body mass (M) as M^0.33, area M^0.67, and volume M^1. From these geometric relationships the diffusive and advective conductance of the spiracles were estimated using reasonable simplifying assumptions. Clearly, diffusion cannot deliver sufficient oxygen to sustain the metabolic requirements of large or active beetles, and the bulk flow of air must be required. Given that the advective conductance scales with M^1.1, this mechanism scales with a sufficiently high exponent meet the oxygen delivery needs of active insects of all masses.Comparing the scaling exponents for spiracle conductance against an exponent of 0.75 for insect metabolic rate is potentially an issue, since the capacity of the respiratory system and its spiracles should be matched to the insect's maximum aerobic O2 uptake requirements (i.e., during flight or activity) not their metabolic rate at rest when oxygen demand is low. The metabolic rate of flying insects scale with body mass with an exponent that is substantially higher than M^0.75, with interspecific analysis suggesting M^1.10 (Niven and Scharlemann 2005) or intraspecifically M^1.02 in hopping locusts (Snelling et al. 2011). Interestingly, the flight exponent is also the same exponent calculated in this study for the scaling of spiracle advective conductance (M^1.1). As such, estimating the metabolic rate of flying endothermic beetles to be 90× resting MR, while also assuming resting metabolic rate scales with mass^0.75, doesn't capture the iso/hyperallometric relationship between body mass and metabolic rate during flight or vigorous locomotory activity, but the scaling exponent for advective conductance is comparable to the hyperallometric value for flight metabolic rate.

We agree that there is some evidence that flight metabolic rate scales with a higher exponent than 0.75 in insects, and we have added this point to the text. We have also included a reference to a new, recently accepted manuscript from our lab (Duell and Harrison, manuscript included as a supplementary file in this submission) which found that flight metabolic rate scales hypermetrically (slope = 1.19) for insects below 58 mg, and hypometrically (slope = 0.67) above 58 mg. This recent paper utilized all of the data in the Niven and Scharlemann’s analysis (n = 54) combined with newer data for 38 additional species collected in our lab and others. In the Duell and Harrison study, the regression line for insects larger than 58 mg (which would include all of the beetles used in this manuscript) had a slope significantly lower than 1, supporting our conclusion that advective capacities scale with a steeper slope than flight metabolic rates, at least over this range of body masses. In any case, our primary conclusion that diffusion though the spiracles becomes increasingly challenging with larger insects remains unchanged. Spiracular advective capacities (which scale 1.1 with mass) match the scaling of flight metabolic rate if Niven and Scharlemann’s analysis is used, or exceed the scaling of flight metabolic rates if the Duell and Harrison analysis is used.

An ideal resolution of this issue would require measurement of the flight metabolic rates of a large number of scarab beetles, as suggested by reviewer 2, as it is plausible that the scaling of flight metabolic rate is order-specific. Unfortunately, most scarab beetles cannot hover or even sustain flight in containers small enough allow detectable changes in carbon dioxide levels, and so it is currently impossible to measure their free-flight flight metabolic rates by gas exchange. We developed a brain stimulation method that induced strong flight in tethered flying Mecynorrhina savagei, one of the large beetles used in this study, and were able to measure flight metabolic rate for this species; the data from which is included in the Duell and Harrison study. Because of these technical challenges, measuring the scaling of flight metabolic rates for the beetles used in this study is beyond the scope of this manuscript. We examined the scaling relationship for the four beetle species for which there are published records, plus our data for M. savagei, and, not surprisingly, the confidence limits for the calculated regression line included 0.67 and 1.2, so this approach was not able to resolve this issue. We have added information to the text to describe the uncertainties of how flight metabolic rate scales in insects, and beetles in particular, and have redone Figure 3c to show how the required P_O2_ gradients across the spiracles will vary if flight metabolic rates scale with exponents of 0.67 or 1.19. We now conclude that spiracular advective capacities may match or exceed flight gas exchange needs.

An implicit assumption in this analysis is that how the spiracles scale with body mass reflects the oxygen delivery capacity of the insect's entire respiratory system. This would be true if either every part of the insect's respiratory system has a conductance that is matched with every other part (symmorphic), such that every part of the gas exchange pathway shared the same scaling exponent and equal capacity for oxygen delivery. If not, then the spiracles must possess the lowest (i.e., rate-limiting) conductance for oxygen uptake – the rest of the tracheal system could have an overall higher capacity or a higher scaling exponent, but this capacity would be unrealized due to the restricted transport of oxygen through the spiracles. In either of these cases, the scaling relationship of the spiracles conductance with body mass would be the same as the oxygen delivery capacity of the insect's tracheal system. However, it is also possible that spiracular conductance scales with a higher exponent than the conductance of other parts of the respiratory system, in which case the spiracles might not be the rate-limiting part of the oxygen transport pathway, and would possess an unrealized capacity. In this case, the scaling exponent for the spiracles would not reflect the tracheal system's oxygen delivery capacity. The validity of the authors' choice in assuming that the insect respiratory system is symmorphic (or that the spiracles are rate-limiting) is partly addressed with the observation that the metabolic rate of *Drosophila* is reduced when a single thoracic spiracle is blocked (L124). But is there any other evidence to support symmorphosis or rate-limiting spiracles? Given that this relationship must be true to accept the overall argument being presented, it would be good to see additional arguments put forward to support this position.

This comment from the reviewer made us realize that we needed to work to clarify our manuscript to indicate that all of our calculations and discussions refer only to transport through the spiracles, not the entire tracheal system. We had already used the adjective “spiracular” to indicate that diffusive and advective capacities only indicate transport through the spiracles, not the entire tracheal system. But we have now added sentences at several points in the manuscript to emphasize this point. It is certainly plausible that the entire tracheal system scales similarly as we have found for spiracles, but our data do not address that.

We have added information to the discussion in an attempt to make it clear that this study does not resolve the question of which steps in gas exchange likely have the greatest relative resistance, or how the overall tracheal system scales in insects.

How the conductance of the spiracles is related to the insect's total oxygen uptake is even more complicated when considering advection, since oxygen delivery through the tracheal system isn't limited by atmospheric oxygen partial pressure as diffusion rate is, and can be increased dramatically, essentially only being limited by the capacity of the tracheal air-sacs' ability to pump air. However, how the tracheal pump's power (Power = Pressure x Flow or 'Q') scales with insect body mass may not yet be known. Without knowing this, there is some uncertainty in predicting what the oxygen delivery associated with the hyper-allometric scaling of spiracular advective conductance would be.

Absolutely. We agree, and have incorporated some of the reviewer’s text into the discussion. Our data can be used to estimate diffusion gradients that will occur across spiracles if diffusion is the sole mechanism for gas exchange (which is likely in some cases, for example during recovery from drowning). Our data also allows us to calculate spiracular resistance to flow, but we completely agree that scaling of advection in insects will require direct measurements of advective flow. As the reviewer knows, this is considerably more challenging than in vertebrates due to the multiple entry and exit points for the respiratory system, and advective flow only been measured for a few insects, especially during flight, so this remains a future goal for the field.

While diffusion can operate through all spiracles simultaneously, the total advective capacity can vary depending on which spiracles act as influx and efflux points (assuming continuous advective flow), as well as how these elements are interconnected, or whether all spiracles function together simultaneously during a period in inhalation followed by exhalation. Thus, the total advective capacity and the resulting oxygen delivery rate is determined by how the spiracles operate together. In this paper the total advective capacity is assumed to be the capacity of all spiracles functioning simultaneously (i.e., the summed capacity of all sixteen spiracles) which would only be possible for half the time (assuming inhalation and exhalation are of equal duration).

We also agree with this important point, and have added sentences to the discussion as suggested.

Overall, this paper does achieve its aim of generating a valuable data set and using this to determine how spiracle morphology scales with beetle body mass. The analysis presented convincingly shows that while diffusion could support the metabolic oxygen demands of a small or resting beetle, advection is required to deliver the oxygen needed for any energetic activity. This also suggests that an insect's size is not constrained by spiracular gas exchange even if the spiracles grow proportionally with body mass and insect size. From an evolutionary point of view this is interesting as it suggests prehistoric giant insects would likely have conformed to this pattern, indicating their size was not limited by oxygen delivery capacity.

We agree that our data on the scaling of spiracular advective capacities provides some support that prehistoric giant insects would likely not have been limited by spiracular oxygen transport, though as noted by the reviewer above, this analysis does not include the oxygen transport capacity of the entire system. We have added a few sentences to the discussion on this point.

While the morphometrics and scaling exponents that are derived from them are all very nicely done and very clear, I think that there needs to be some discussion to explain the rationale behind your use of the M^0.75 relationship for MR as the exponent you are comparing the spiracular conductance exponents against, rather than using scaling exponents derived for flying or active MR, where the exponent is >1.

As noted above, we have added new information, based on a recently accepted manuscript, on the scaling of flight metabolic rates in insects, and redone Figure 3b to predict the P_O2_ gradient across the spiracles depending on two “extreme” estimates of the scaling of flight metabolic rate. We have also added material discussing the uncertainties in the scaling of flight metabolic rate in insects, and beetles in particular.

Likewise, I'd be keen to see some mention of the assumptions underlying why examining the diffusive and advective conductances of the spiracles is revealing, when the capacity of this comparatively small part of the gas exchange pathway may exceed the internal conductance. Ideally more compelling evidence should be provided showing either symmorphosis of gas transport across the tracheal system or that the spiracles are likely to be the rate-limiting conductance within the tracheal system.

Our data do not address whether symmorphosis occurs in the tracheal system, or the relative importance of spiracular resistance to gas exchange. Doing so would require measures of the other aspects of the tracheal system, a nontrivial endeavor, and ideally assessment of the relative roles of diffusion and advection in the various steps (for which we currently lack published methods). Nonetheless, we would argue that the measurements we have provided of the spiracles are novel and do provide important fresh insights into the evolution of the insect tracheal system. We show, for the first time, that spiracles scale isometrically, and that this means that diffusion across the spiracles becomes increasingly more challenging as insects increase in size. We also make the important and, to our knowledge, new point that this relationship between isometric scaling of respiratory structure and diffusive capacities should be general for animals. We have worked to revise the discussion and abstract to ensure that this point is clear.

Likewise, it'd be great to see some rationale behind how you might expect the tracheal pump's capacity to scale with body mass, since this will determine if the oxygen delivery capacity associated with the spiracle's advective conductance also scales with M^1.1. For example, if it is assumed that tracheal pump power scales isometrically (M^1), then as advective conductance scales with M^1.1, would this increase flow, and therefore oxygen delivery, with the same exponent? Would pressure decrease in larger insects? Presenting some background to the assumptions around how the insect generates an advective flow through its spiracles, and how this might scale with insect body mass, is important to be able to appreciate how increasing spiracle conductance would change the volumetric flow of air and, therefore, oxygen delivery.

Unfortunately, there are no literature data on how advection scales in insects, and, as noted above, measuring advective flow, especially during flight, is nontrivial. We have added some material to the discussion on this topic to clarify this point.

It is interesting that the mesothoracic spiracles show the tightest relationship with bodymass, given that these spiracles lie closest to the most metabolically demanding tissue: the thoracic flight musculature. Given the possibility/likelihood of unidirectional advective flow during activity (in through the thoracic and out through the abdominal spiracles), how does the summed advective conductance of the thoracic spiracles compare to that of the summed abdominal spiracular conductance? Is there an excess advective capacity in the abdominal spiracles, assuming they are functioning as "exhaust spiracles" relative to the thoracic "intake spiracles"? Would assuming continuous unidirectional flow (in through some spiracles and out through others) alter the scaling exponent or only the elevation of the advective conductance relationship? Could this be considered in the analysis?

This is a very interesting point that we had not thought of. In fact, the summed conductances of the thoracic spiracles far exceed the conductances of the summed abdominal spiracles; we have added new supplementary tables that enable a reader to see this easily. The mechanisms of gas exchange in some flying beetles were addressed by Miller (1966), and we have used information from his analysis to address these interesting points in the discussion.

Specific comments:L33: I'd consider changing the exponent you consider from the resting metabolic rate (M^0.75) to that for flight MR (M^1.1)

As noted above, we feel that the best, newest data indicates that flight metabolic rate scales hypometrically (slope = 0.67), but have added discussion of the uncertainties in how flight metabolic rate scales in insects and beetles in particular.

L114: "… remains unclear how the components of the system scales". Change to "scale"

Done.

L266: "The mesothoracic spiracle was" change to "were"? I know you only measured one, but there are two of them

Done.

L294: "one spiracle scales isometrically" change to "one spiracle pair scales isometrically"

Done.

Reviewer #2 (Recommendations for the authors):The study aimed to determine gas transport capacity of tracheal spiracles in different sized scarab beetles using micro-CT scans. The authors assumed that metabolic rate scales with a scaling exponent of 0.75. They found that spiracle size does not sufficiently increase with increasing body size to allow diffusive oxygen supply but increases more than required to satisfy metabolic demands during advective gas exchange. The data are of interest for Biologists working on the respiratory system of animals but need experimental proof of the scaling exponent used as a reference.The entire conclusion of the study is based upon the assumption that metabolic rate exactly scales with a 0.75 exponent.

We disagree. We demonstrate isometric scaling of the spiracles, and show 0.33 that this means that diffusing capacities of the spiracles scale with m. Our major conclusion is that diffusion across the spiracles becomes increasingly challenging as beetles become larger. That conclusion would only be invalid if metabolic rates scaled interspecifically with a scaling exponent of 0.33 or less. The scaling of metabolic rate has been documented many times for insects, and the exponents are always considerably higher than 0.33, using in the range of 0.75. Our primary conclusion is robust and not dependent on any exact scaling exponent of metabolic rate.

Many previous studies, however, showed that this scaling exponent is only valid among a large range of body sizes and (to some extent) including also vertebrates. In single clades, scaling exponents may significantly be different from 0.75. This means that the finding that diffusion is not sufficient in larger beetles depends on the correct scaling coefficient for metabolic rate in these animals. The authors do not provide separate measurements of metabolic rate to more reliably estimate the 0.75 coefficient in scarab beetles. This is, however, critical for the outcome of the study.

We addressed this issue in detail in the response to reviewer 1. Briefly, we have added data from a recently accepted manuscript that supports our prior suggestion that flight metabolic rates scale hypometrically in large insects (with a slope of 0.67), very similar to the scaling pattern that has been shown for flying birds and bats. We agree that it would be wonderful to have flight metabolic rates for a larger number of scarab beetles; unfortunately, as described above, this is a very challenging proposition and so is beyond the scope of this paper. Accordingly, we have revised one aspect of our conclusions, and now indicate that spiracular advective capacities may exceed or match the scaling of flight metabolic rates.

In equations 1 and 2, the authors nicely explain that diffusion should linearly depend on spiracle geometry. This assumption matches the data in figure 2, showing slopes close to 0.75. In figure 3A, by contrast, total diffusive capacity increases much less than spiracle geometry, which runs apparently counter to the data in Figure 2. This needs an explanation.

Figure 2 shows that spiracular area scales approximately with mass 0.67, and spiracular depth scales approximately with mass 0.33. Spiracular diffusing capacity, which is shown in Figure 3, depends on spiracular area divided by spiracular depth (equation 1). 0.67 – 0.33 = 0.34, which is approximately the scaling of total spiracular conductance as shown in Figure 3.

The authors leave open the question of how important spiracle opening area is for oxygen flux compared to the rest of the tracheal system. Even assuming that spiracle area satisfies oxygen supply via diffusion, an animal might rely on advective flow because of other tracheal constraints. The above concern also holds for the slope assuming advective oxygen supply. For very small beetles, moreover, equations 3 and 4 might be too simplistic because they do not consider the fluid mechanic effects associated to flows at low Reynolds number. While Reynolds number-dependent phenomena do not change much at large Reynolds number, the thick boundary layer might hinder advective flow at low Reynolds numbers.

As pointed out above, our data are restricted to spiracular morphology, and our analysis is based on how body size affects the diffusing and advective capacities of the spiracles; we did not measure and cannot draw conclusions about the remainder of the tracheal system. We agree that small insects can and do use advection, even at rest. The fact that this occurs is one of the major unresolved issues in insect respiratory physiology. We have added discussion of this interesting point to the discussion.

The study determined gas transport capacity of tracheal spiracles in different sized beetles using micro-CT scans. The authors found that spiracle area does not sufficiently increase with increasing body size for diffusive oxygen supply. Assuming advection, by contrast, spiracle area increases more than required to satisfy metabolic demands. The manuscript is written clearly and the topic is of interest for Biologists working on the respiratory system of animals. Although I much sympathize with the approach and the data, my impression is that findings and conclusion are too controversial and thus recommend publication in a more specialized journal.The authors only compare their findings to the 0.75 slope at resting metabolic rate. On page 7, however, they mention that spiracle morphology should match gas exchange needs at peak metabolic performance. I assume that all tested species are capable of flight (?). As flight costs increase with decreasing body size due to viscous drag on wings and body, we would not expect isometric scaling of spiracle openings for diffusive gas exchange. This aspect should be considered in a revised version of the manuscript.

This topic is discussed in detail in response to reviewer 1. Briefly here, all beetles used here are capable of flight, but do not hover, and so measurement of flight metabolic rates by gas exchange poses technical challenges in these species that are beyond the scope of this study. We now have added a paragraph to the discussion explaining the uncertainties in the scaling of flight metabolic rates for these beetles. We have added new references from a recently accepted manuscript that documents that flight metabolic rates scale differently in insects depending on body size. For small insects (below 58 mg), flight metabolic rates scale hypermetrically, plausibly because of the changes that occur in aerodynamic requirements as Reynolds number changes. For flying insects heavier than 58 mg, flight metabolic rates scale hypometrically, as observed in vertebrates. To address the concerns about uncertainty in the scaling flight metabolic rates, we have revised our conclusions to indicate that spiracular advective capacities either match or exceed gas exchange needs during flight.

Length. The manuscript consists of 5 pages Introduction, 7 pages Methods, 1 page Results and 4 pages Discussion sections and thus needs a major revision towards balanced section length. The data set is comparatively small

While the number of species is comparatively small, the morphological detail in the measurements of spiracular morphology is large.

Approximately one person-year was required to collect all of the micro-CT images, and an additional person-year to analyze all of them. No prior studies have presented similar detailed morphological analyses of spiracular diffusing capacity, even for a single insect species or spiracle. Thus, the assessment of all spiracles for ten species, including some of the largest insects alive today, is a major advance. As noted in the introduction, although invertebrates represent the majority of all species and are of tremendous ecological and evolutionary importance, we lack fundamental data on how body size affects respiratory system structure and function, and so feel that the material we present here is a major advance. Finally, we make a new point, to our knowledge, that isometric scaling of respiratory structures will be associated with reduced mass specific diffusive capacities generally in animals.

and I suggest to add measurements of metabolic rates for each beetle (see comment above).

As noted above, measurement of flight metabolic rates of flying beetles is technically very challenging because most cannot hover; so unfortunately, this suggestion is not practical. As noted above, we have added a reference to a new paper whose analysis supports the idea that flight metabolic rates in large insects scale hypometrically, as in vertebrates.

Statistics. The data in figures 2 and 3 are barely normally distributed and my impression is that the slopes thus strongly depend on the two data points of the smallest beetles (-1 body mass). As the slope only depends on 10 data points in total, I recommend further statistics that evaluates the unequal(?) data distribution.

We also had similar concerns, which is one reason we opted to perform our analysis in both a Bayesian parametric way (with clearly stated assumptions of normality) and frequentist non-parametric way (which doesn’t make assumptions of normality).

Non-parametric approach – no assumption of normality – takes into account the uneven data distribution

Our non-parametric bootstrapping approach does not rely on an assumption of normality to generate confidence intervals on the slopes. In particular, the bootstrapping operates by taking a random sample from the dataset (with replacement) of the same number of datapoints as the original dataset and calculating a summary statistic on that sample (for our inference, the summary statistic is a slope generated by numerical optimization). On average, around 11% of the bootstrap sample generated by this procedure and which we calculated slopes for do not contain either of the small beetles as datapoints. Thus, the results of the absence of these points influences the range of slopes generated and hence the bootstrap derived confidence intervals. In figure 2B, for example, the confidence interval for the range of slope estimates is non-symmetric, with a larger range of low slopes compared to the median as opposed to slopes higher than the median. Bootstrap samples with no small beetles are one reason for this larger range of smaller slopes in the posterior spiracles. Further, for fitting the regressions whose slopes we used as a summary statistic we used a minimization of the sum of squares of the residuals via numerical optimization; this approach doesn’t require or assume a normal distribution about a line for the data. It calculates the best fit via optimization as mentioned. To get an estimate on the ‘σ’ from the bootstrapping used in figure 2C to look at how variable the different spiracle sizes were, we calculate the residual standard deviation for each regression output from our bootstrap sample; calling this ‘σ’ does analogize it the σ of a normal distribution (it is the standard deviation of a normal distribution if the data are normally distributed about the fit line). Regardless of whether the data are normal, these residual standard deviations give an indication of how far away the datapoints are on average from the fit line and, we think, are worth displaying. The results of estimates for σ of an explicitly normal Bayesian model are also shown and show the same trend as the residual standard deviations from bootstrapping. The non-parametric confidence intervals for 2C are also generated by bootstrapping and so take into account what the absence of the small beetles does to the parameter estimates.

Almost all of our plots in the main text which show regressions/confidence intervals (except figure 2C,F) use the non-parametric analysis described above which takes into account both the sparse data distribution for the small animal body sizes and doesn’t assume normality.

Parametric approach

We agree that, especially given the small number of datapoints, it is difficult to have strong confidence that the datapoints are normally distributed about the linear mathematical model that we estimate parameters for. Even so, we wanted to perform our analysis in multiple ways so we could use separate statistical techniques to have more confidence in the robustness of the parameter estimates we generated. Hence, we made a Bayesian model which explicitly assumes that our data points are generated by a normal distribution with mean on a line with parameters for slope and intercept and a standard deviation. The posterior predictive distributions for the resulting models do look like the normal model does a pretty good job of encapsulating the data (supplemental figure 4, the squiggly grey intervals represent 80% and 95% percentiles for data drawn from the posterior for the model). For these posterior distributions, generally 1-2 out of 10 datapoints falls outside the 80% confidence interval, which is about right given the point numbers. However, due to concerns that the normal distribution was too strong an assumption, we ran with the above-described non-parametric approach for almost all purposes of inference.